# Neuroprotective Effects of the Psychoactive Compound Biatractylolide (BD) in Alzheimer’s Disease

**DOI:** 10.3390/molecules27238294

**Published:** 2022-11-28

**Authors:** Qianmei Hu, Jixiang Wang, Maida Irshad, Siyu Mao, Hongting Chen, Yujiao Song, Xuan Xu, Xing Feng

**Affiliations:** 1The Key Laboratory of Study and Discovery of Small Targeted Molecules of Hunan Province, and Department of Pharmacy, School of Medicine, Hunan Normal University, Changsha 410013, China; 2College of Letters & Science, University of California, Berkeley, CA 94720, USA

**Keywords:** traditional Chinese medicine, biatractylolide, neuroprotective, pharmacological model, network pharmacology

## Abstract

Mitochondria play a central role in the survival or death of neuronal cells, and they are regulators of energy metabolism and cell death pathways. Many studies support the role of mitochondrial dysfunction and oxidative damage in the pathogenesis of Alzheimer’s disease. Biatractylolide (BD) is a kind of internal symmetry double sesquiterpene novel ester compound isolated from the Chinese medicinal plant Baizhu, has neuroprotective effects in Alzheimer’s disease. We developed a systematic pharmacological model based on chemical pharmacokinetic and pharmacological data to identify potential compounds and targets of Baizhu. The neuroprotective effects of BD in PC12 (rat adrenal pheochromocytoma cells) and SH-SY5Y (human bone marrow neuroblastoma cells) were evaluated by in vitro experiments. Based on the predicted results, we selected 18 active compounds, which were associated with 20 potential targets and 22 signaling pathways. Compound-target, target-disease and target-pathway networks were constructed using Cytoscape 3.2.1. And verified by in vitro experiments that BD could inhibit Aβ by reducing oxidative stress and decreasing CytC release induced mPTP opening. This study provides a theoretical basis for the development of BD as an anti-Alzheimer’s disease drug.

## 1. Introduction

Alzheimer’s disease (AD) is a progressive neurodegenerative disease in the elderly [1,2]. This disease causes severe cognitive impairment and is accompanied by classic pathological changes. Such as amyloid beta (Aβ) deposition in the brain parenchyma, blood vessels [3,4,5], formation of neurofibrillary tangles [6,7,8,9], and loss of neurons [10]. Clinically, the silent period of the disease may last for decades [11], with some symptoms lasting 5 to 10 years [12,13].

Aβ plaques, or possibly soluble oligomers of Aβ, are neurotoxic and act by disrupting mitochondrial function, inducing apoptosis, forming ion channels and stimulating stress-activated protein kinase pathways [14]. It has been shown that Aβ-mediated mitochondrial dysfunction is due to the activated mitochondrial permeability transition pore (mPTP), which increases mitochondrial membrane permeability [15,16]. The mPTP is known to consist of voltage-dependent anion channels, adenine nucleotide transporters and cell cycle protein D (CypD) [17,18]. CypD is critical in stabilising the mitochondrial permeability transition which acts to switch on mPTP. Aβ can accumulate in mitochondria, in which it interacts with ABAD and CypD [19,20,21], this interaction promotes ROS leakage and ultimately leads to mitochondrial dysfunction [22,23,24,25]. Increased permeability of the mitochondrial membrane after the opening of mPTP [15] leads to the rupture of the outer mitochondrial membrane, followed by the diffusion of contents such as calcium and CytC from the mitochondrial matrix into the cytoplasm, where they eventually trigger cell death [26,27,28]. Thus Aβ impairs mPTP function by disrupting the mitochondrial membrane potential [29] and increasing ROS production, mitochondrial swelling and CytC release [30,31,32], ultimately leading to cell death.

A number of approaches have been explored to protect mitochondria from Aβ-induced damage, including elimination of ROS [33] enhancement of clearance [34,35] and stabilization of calcium homeostasis [36]. Research that block the formation of large amounts of mPTP are advantageous compared to strategies that focus primarily on injury prevention by eliminating damage-causing factors [37,38].In the search for more novel therapeutic agents, the laboratory focused on traditional Chinese herbs.

Baizhu is a traditional Chinese herb with roots possessing various pharmacological activities such as antioxidant [39], gastric protection [40], antitumour [41] and anti-AD [42,43].Biatractylolide (BD) is a novel bisesquiterpene lactone isolated from the Chinese medicinal plant Atractylodes macrocephala (Baizhu). According to previous studies, BD has a pronounced effect on reducing acetylcholinesterase activity and improving memory in mice with aluminium trichloride induced dementia. It could significantly reduce acetylcholinesterase activity in a rat AD model induced by Aβ_1-40_ and improve behavior and memory. Therefore, BD showed promising potential to be an effective anti-AD drug [44].

The leading model of “one gene, one target, one disease” has been found to influence drug discovery [45], with many effective drugs working on multiple targets rather than a single target. Thus, network pharmacology provides a new network model for multiple targets, multiple effects, and complicated diseases [46,47].

In the present study, we developed a systems pharmacology-based model to screen potential compounds and targets and applied compound-target (C-T) and target-disease (T-D) networks to assess the mechanism of action. The neuroprotective effect of BD against Aβ-induced oxidative stress was assessed by in vitro experiments using Aβ_25-35_ [48] peptide in rat adrenal pheochromocytoma cells (PC12) and human myeloid neuroblastoma cell line (SH-SY5Y) to mimic Aβ-induced neurodegeneration.

## 2. Results

### 2.1. MTT Assay

The MTT assay was used in order to investigate the therapeutic effect of BD on Aβ_25-35_-induced cell damage (Figure 1). The cell viability treated with Aβ_25-35_ was significantly reduced. After pretreatment with different concentrations of BD (5 μM, 10 μM and 20 μM, respectively) for two hours followed by treatment with Aβ_25-35_, the cell viability of SH-SY5Y cells was significantly increased. Cell viability increased to 90.5 ± 0.3% at a BD concentration of 20 μM. After pretreatment with BD, the cell viability of PC12 cells also increased significantly, increasing to 82.2 ± 1.4% at a BD concentration of 20 μM.This result suggests that BD has a significant effect on improving cell viability in both cell.

### 2.2. Detection of MMP by Rhodamine 123 Staining

After treatment with 20 µM Aβ_25-35_, MMP was considerably decreased as compared with the control (*p* < 0.001) (Figure 2). Pretreatment with BD (5, 10, and 20 μM) could significantly protect PC12 and SH-SY5Y cells from the effects of reduced MMP by A_25-35_. In PC12 cells, the MMP increased to 76.7 ± 2.4%, 84.9 ± 1.0%, and 91.6 ± 0.7%, respectively, after BD (5 μM, 10 μM and 20 μM, respectively) pretreatment. In SH-SY5Y cells, MMP increased to 78.7 ± 2.0%, 81.8 ± 1.0%, 86.8 ± 1.2%, respectively, after BD (5 μM, 10 μM and 20 μM, respectively) pretreatment.

### 2.3. Measurement of Intracellular ROS

The effects of different concentrations of BD on ROS release in cells were investigated (Figure 3). With treatment, relative fluorescence of ROS was increased in SH-SY5Y and PC12 cells as compared to controls. Pretreatment with BD at 5, 10, 20 μM, however, significantly inhibited relative fluorescence intensity of ROS to 149.5 ± 4.5%, 118.5 ± 2.5%, 112.5 ± 6.5% in PC12 cells as compared with Aβ_25-35_ treatment (*p* < 0.01). In addition, BD (5, 10, and 20 μM) gradually decreased ROS production to 174.5 ± 4.5%, 139 ± 1.0%, 119 ± 2.0% in SH-SY5Y cells as compared with Aβ_25-35_ treatment.

### 2.4. Protein Characterization

For mPTP opening, CypD is an important factor that combines with VDAC and ANT and forms mPTP.The opening of mPTP causes the release of the pro-apoptotic factor CytC from the mitochondrial matrix to the cytoplasm. To validate the protective effects of BD on Aβ_25-35_–induced PC12 and SH-SY5Y cells, we further analyzed protein characterization by western blot analysis. BD treatment could reduce Aβ_25-35_–induced CytC release (Figure 4). Aβ_25-35_ treatment significantly increased CytC in cytosol (* *p* < 0.05) while after BD treatment the level of CytC was significantly reduced (*p* < 0.01).

### 2.5. Networks

#### 2.5.1. Active Compound Screening

We screened 18 active compounds with their 388 potential targets by using OB ≥ 25%, DL ≥ 0.1, Caco-2 cell permeability ≥ 0.4. Furthermore, we screened 20 targets that are related to most compounds (Figure 5).

#### 2.5.2. Network Construction and Analysis: C-T Network

We build a C-T network based on the candidate compounds of Baizhu and potential targets. The C-T network included 18 nodes (18 candidate compounds) and 20 edges (20 latent targets) (Figure 6).

#### 2.5.3. T-D Network

To better comprehend the diseases modified by Baizhu, we searched the Drug Bank and TTD databases for latent targets to find corresponding diseases. The 78 diseases were categorized into 10 groups on the grounds of MeSH Browser (2014 version). And the T-D network was structured by potential targets and their corresponding diseases. Lots of defined diseases belong to the diseases of central nervous system (24/78), bone and joint (14/78) or digestive system (13/78) (Figure 7). For instance, PTGS2, MAPT, and GABRA1 are critical targets of leukodystrophy and are associated with AD, neuropathic pain, and cognitive impairment in the T-D network (Figure 7).

#### 2.5.4. Target-Pathway (T-P) Network

To explore the regulation of Baizhu for treating AD, pathways were assembled by using Cytoscape (Figure 8) based on current knowledge of pathogenesis. The potential human target proteins were searched in the KEGG pathway and GO databases. KEGG pathways included oxidative, metabolic, calcium signaling, and PI3K-Akt signaling pathways. These pathways are related to AD. According to the pathways we found, Baizhu could alleviate the severity of AD by their inhibition (↓ROS).

## 3. Discussion

In this study, we developed a systematic pharmacological model based on chemical, pharmacokinetic and pharmacological data to identify potential compounds and targets of the traditional Chinese medicine Atractylodes macrocephala with AD neuroprotective effects. We selected 18 active compounds associated with 20 potential targets and 22 signaling pathways by OB screening and DL and Caco-2 cell permeability evaluation, and successfully constructed C-T, T-D and target pathway networks using Cytoscape 3.2.1. We also demonstrated in cellular experiments that BD can inhibit the effects of Aβ_25-35_-induced cell damage and inhibit the reduction of MMP, BD can reduce the production of ROS in SH-SY5Y cells, and BD treatment can reduce Aβ_25-35_-induced CytC release by WB experiments. BD is a promising novel compound for the treatment of Alzheimer’s disease.

At present, due to the aging of the population, the prevalence of AD is increasing, putting a very high pressure on society and families [49,50]. However, the pathogenesis of AD has not been clearly interpreted and valid therapy is still lacking. Aβ is one of the hallmarks [51] of MMP. Aβ itself causes oxidative stress but also increases the oxidative stress produced by others that leads to mPTP opening. mPTP, when formed, constitutes a non-selective, highly conductive pore that allows calcium transport and also transport of any solute below the pore size. This leads to mitochondrial osmotic swelling and MMP dissipation, impairing the mitochondrial respiratory chain, resulting in decreased adenosine triphosphate production and increased ROS production. Second, mitochondrial swelling result in rupture of the outer membrane, allowing the release of apoptogenic factors such as CytC from the mitochondria to the cell membrane (Figure 9).

Many approaches have been investigated to protect mitochondria against Aβ-induced injury, such as eliminating ROS [33], strengthening Aβ clearance [34,35] and regulating calcium balance [36,37]. Strategies that focus on preventing mitochondrial insult by blocking large amounts of mPTP formation are advantageous because they can improve mitochondrial resistance to existing damage. Mitochondrial targeted therapies that ameliorate mitochondrial function may hold great promise in preventing and treating AD.

Chinese herbal medicine has been considered one of the most momentous tactics in complementary and alternative medicine. Also, network pharmacology has become a effective tool to comprehend drug targeting, particularly for the various components of Chinese herbal medicine. From Baizhu (Chinese herbal medicine) via ADME prescreening (TCMSP), we identified 18 candidate compounds with 388 (Swiss Target Prediction) potential targets possessing favorable pharmacokinetic profiles. Then we focused on 20 targets that are related to most compounds. We created three networks: (1) C-T network, with effective constituent of Baizhu and its corresponding targets used (TCMSP, Swiss Target Prediction); (2) T-D network, with all targets and their corresponding diseases used (TTD, Drug bank); and (3) target-pathway network (T-P network), using target information extracted from the KEGG database and the T-P network constructed from targets and their corresponding assumed pathways. All visualized network diagrams were built with Cytoscape 3.2.1 (http://www.cytoscape.org/, accessed on 3 November 2022), an open software package project for visualizing, integrating, modeling and analyzing interaction networks [52].

From the results, the Chinese herbal medicine Baizhu may have a protective effect on CNS and other systems. BD (M62) is one ingredient of Baizhu that has a protective effect in AD [43]. It has antioxidant activity, inhibits the opening of mPTP and decreases MPT-dependent CytC release. So it could have a neuroprotective effect in AD. Furthermore, we used PC12 and SH-SY5Y cell lines and found increased cell viability by using preliminary BD against Aβ_25-35_–damaged cells.

## 4. Materials and Methods

### 4.1. Materials

BD was isolated from the ethyl acetate extract of A. macrocephala (Baizhu) by multistep chromatographic processing (Figure 10). Aβ_25-35_ and DMSO were from Sigma Chemicals (USA). Antibodies for the proteins CypD, CytC and β-actin were from Protein Tech.

### 4.2. Data Set Construction

We used the literature published in the previous 32 years available in the Traditional Chinese Medicine Systems Pharmacology (TCMSP) database and analysis platform (http://sm.nwsuaf.edu.cn/lsp/tcmsp.php, accessed on 3 November 2022) and the Swiss Target Prediction (http://www.swisstargetprediction.ch/, accessed on 3 November 2022). We searched corresponding targets and A. macrocephala from the TCMSP database.

### 4.3. Active Compound Screening

Initially we searched the Total-55 compounds for Baizhu drug and their corresponding targets. Then we used oral bioavailability (OB) screening with drug-likeness (DL) and Caco-2 cell permeability evaluation to identify the active compounds in Baizhu. We selected 18 compounds with OB ≥ 25%, DL ≥ 0.1, Caco-2 permeability ≥ 0.4 that exhibited extensive [53] pharmacological activities as candidate active compounds for further research.

### 4.4. Network Construction

Compound-target (C-T) and target-disease (T-D) networks according to Cytoscape 3.2.1 were used to identify potential drugs and their corresponding mechanisms using a compound-target-disease association approach. The C-T network was generated by linking screened candidate compounds to potential targets. The T-D network was constructed by linking relevant targets to diseases. Potential diseases were obtained from the Therapeutic Target Database (TTD; http://bidd.nus.edu.sg, accessed on 3 November 2022) and DrugBank (https://www.drugbank.ca/, accessed on 3 November 2022).

### 4.5. Pathway Analysis

Signaling pathways, as an important component of pharmacology, link receptor ligand interactions to pharmacodynamics outputs [54]. First, we obtained the details of the human target proteins and then analyzed their functions and pathways by Gene Ontology (GO) and Kyoto Encyclopedia of Genes and Genomes pathway analysis (KEGG, http://www.genome.jp/kegg/, accessed on 3 November 2022), respectively. Then based on this basic information, pathways related to the central nervous system (CNS; AD) pathology were assembled by using Cytoscape.

### 4.6. Cell Culture

PC12 (rat adrenal pheochromocytoma cell) and SH-SY5Y (human bone marrow neuroblastoma cell line) cells provided by Dr. James R. Woodgett (China Pharmaceutical University, Nanjing, China) were cultured in DMEM medium in the presence of 10% fetal bovine serum and 1% double antibody. All cells were cultured at 37 °C in a humidified 5% CO_2_ incubator.

### 4.7. Experimental Grouping and Drug Treatment

After adding different concentrations of Aβ_25-35_ in the preliminary experiment, by determining dose-response curves, we selected 20 µM (concentration) as the most optimal damage concentration. The experiment was as follows: (1) blank control group, (2) Aβ injury model group and (3) drug treatment group (low, medium and high).

### 4.8. Cell Viability Assay (MTT)

Cells were cultured in 96-well plates for 48 h and then pretreated with different concentrations of BD (5, 10, 20 µM) for two hours before adding Aβ_25-35_ and reacting for 24 h. 50 µL of MTT reagent (3-(4,5-dimethylthiazol-2-yl)-2,5-diphenyltetrazole) was added to each well and incubated for 5 h at 37 °C. Next, the medium was discarded and 150 µL of DMSO per well was added to dissolve the formazan crystals. Cell viability was measured by measuring absorbance at 490 nm using a Dynatech MR5000 plate reader.

### 4.9. Detection of MMP by Rhodamine 123 Staining

The opening of the mPTP in PC12 and SH-SY5Y cells was assessed by using rhodamine 123 staining. PC12 (2 × 10^5^) cells and SH-SY5Y (3 × 10^5^) cells were seeded in 24-well plates and cultured overnight. Then cells were treated with various concentrations of BD (5, 10, 20 μM) for 2 h before Aβ_25-35_ treatment for 24 h in an incubator. Treated cells were harvested and resuspended in the medium, then incubated with rhodamine 123 (10 µg/mL) at 37 °C in a humidified 5% CO_2_ incubator for 10–15 min at room temperature in the dark, then observed by fluorescence microscopy.

### 4.10. Measurement of Intracellular ROS

PC12 cells and SH-SY5Y cells were seeded in 6-well culture plates at 1 × 10^5^/well and cultured overnight. Then cells were treated with 5, 10, and 20 µM BD for 2 h before Aβ treatment for 24 h in an incubator. Next, treated cells were washed with cold PBS twice and fixed with formaldehyde for 30 min according to ROS assay kit instructions (Beyotime Biotechnology). Intracellular ROS contents were observed by fluorescence microscopy.

### 4.11. Western Blot Analysis (CYPD, CytC)

Western blot assays were performed as described by Su et al. [55]. Cellular proteins were extracted, and sample proteins were separated on SDS-polyacrylamide gel electrophoresis and transferred to polyvinylidene difluoride membranes (PVDF). The membranes were incubated with the primary antibody overnight at 4 °C and then incubated with the secondary antibody for 1 h at room temperature. Finally, reaction bands were observed using Pierce Super Signal Chemiluminescent Substrate (Rockford, IL, USA) and proteins were quantified using Image J.

### 4.12. Statistical Analysis

All quantitative data are expressed as mean ± SEM. Statistical differences between the groups were assessed by one-way ANOVA. *p* < 0.05 was considered statistically significant. Statistical analysis involved using GraphPad Prism (GraphPad Software, Inc., San Diego, CA, USA).

## 5. Conclusions

Mitochondrial oxidative injury plays a crucial role in the pathogenesis of multiple neurodegenerative diseases including AD. Hence, approaches to prevent or decrease oxidative damage may offer treatment efficacy. We proofed that biatractylolide overexpression inhibited Aβ-induced mPTP opening by reducing oxidative stress and decreased CytC release. We also used systems pharmacology to understand the pharmacological mechanism and active substances of traditional Chinese medicine. This study defines possible therapeutic targets for AD treatment, although further experiments are needed to support this.

## Figures and Tables

**Figure 1 molecules-27-08294-f001:**
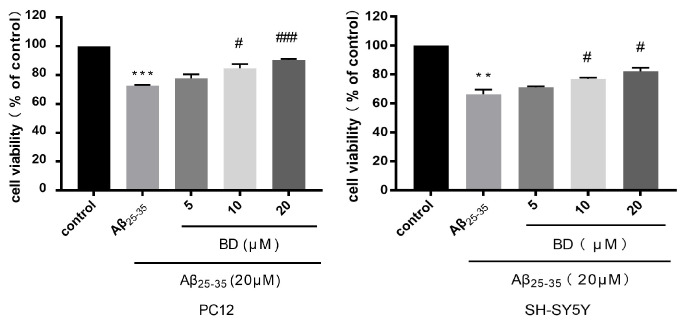
The protective effect of BD on viability of PC12 and SH-SY5Y cells. BD improved cell viability in Aβ_25-35_-treated PC12 and SH-SY5Y cells as revealed by MTT assay. (** *p* < 0.01,*** *p* < 0.005 vs. control ^#^
*p* < 0.05,^###^
*p* < 0.005 vs. treated group).

**Figure 2 molecules-27-08294-f002:**
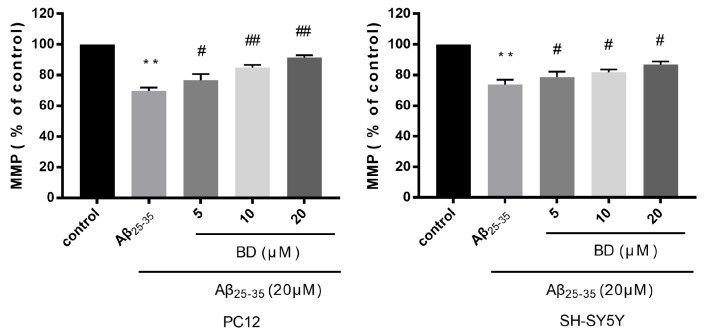
Effect of Aβ_25-35_ mitochondrial membrane potential (MMP) in PC12 and SH-SY5Y cell lines with BD treatment. The two cell lines were treated with low, medium, high, concentrations of BD, then MMP was examined by rhodamine 123 staining. (** *p* < 0.01 vs. control, ^#^
*p* < 0.05, ^##^
*p* < 0.01 vs. treated).

**Figure 3 molecules-27-08294-f003:**
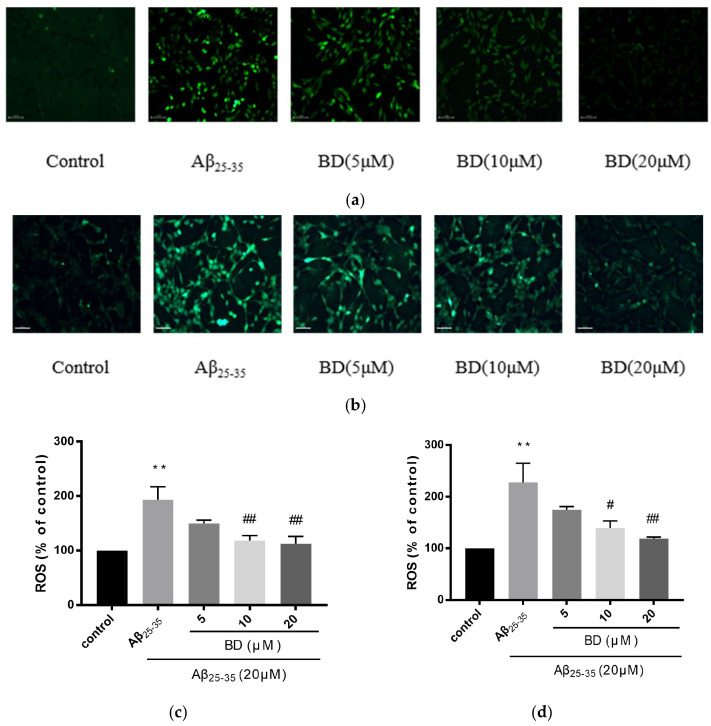
Fluorescence intensity of reactive oxygen species in PC12 and SH-SY5Y cells treated with Aβ_25-35_. Cells were treated with 5 μM, 10 μM and 20 μM BD for 2 h, followed by Aβ_25-35_ for 24 h. (**a**): PC12; (**b**): SH-SY5Y. The bar chart shows the quantitative data. (**c**): PC12; (**d**): SH-SY5Y. (** *p* < 0.01 vs. control, ^#^
*p* < 0.05,^##^
*p* < 0.01 vs. treated).

**Figure 4 molecules-27-08294-f004:**
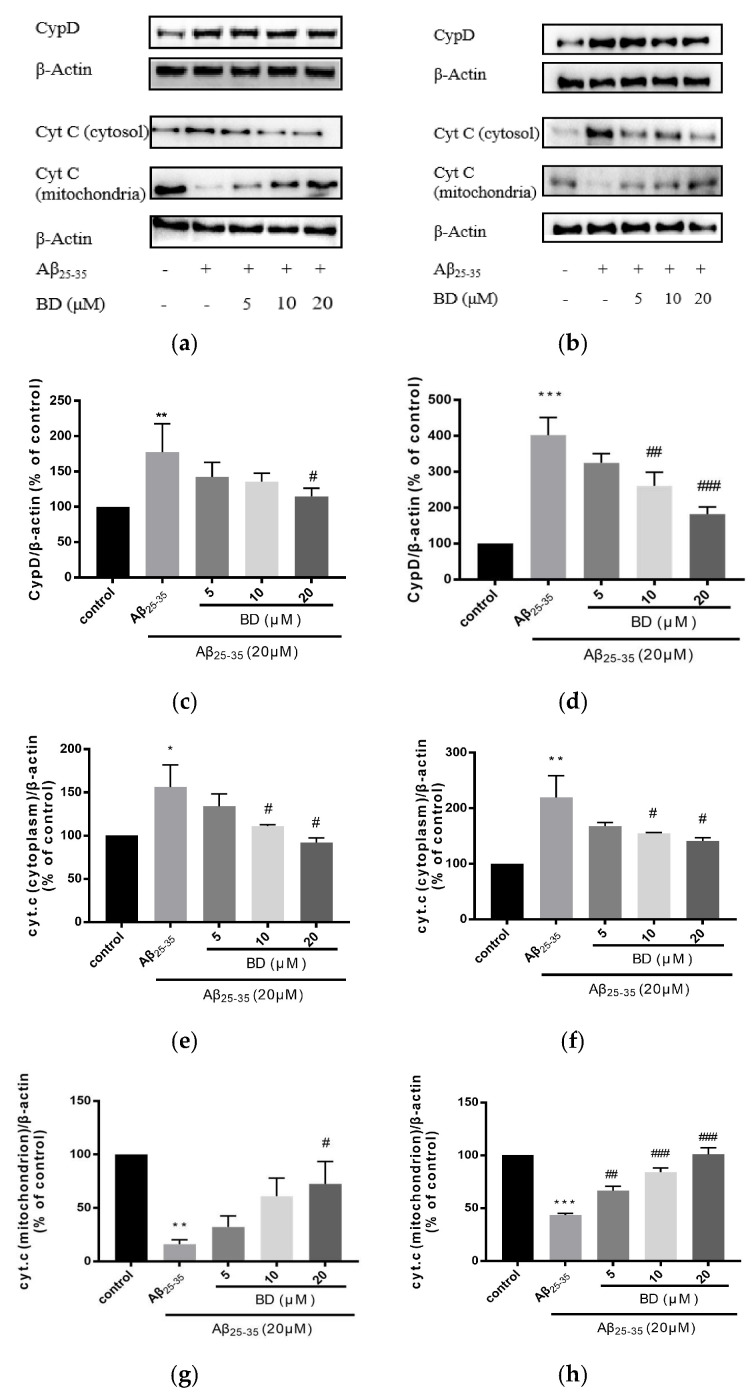
Effects of BD on the expression of CypD and CytC protein in PC12 and SH-SY5Y cells induced by Aβ_25-35_. (**a**): The expression of CypD and CytC protein in PC12 cells; (**b**): CypD and CytC protein expression in SH-SY5Y cells. CypD and CytC were quantified by treating cells with BD at different concentrations for 2 h before treatment with Aβ_25-35_ for 24 h; (**c**,**e**,**g**): CypD and CytC quantification in PC12 cells; (**d**,**f**,**h**): CypD and CytC quantification in SH-SY5Y cells, which is proportional to β-actin level. (* *p* < 0.05, ** *p* < 0.01, *** *p* < 0.005 vs. control. ^#^
*p* < 0.05, ^##^
*p* < 0.01, ^###^
*p* < 0.005 vs. treated).

**Figure 5 molecules-27-08294-f005:**
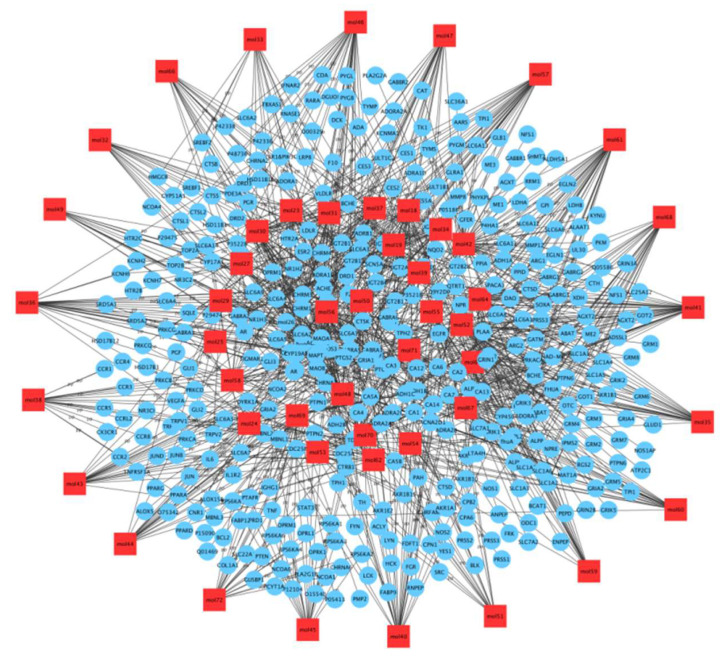
Network of all (55) compounds of Baizhu.

**Figure 6 molecules-27-08294-f006:**
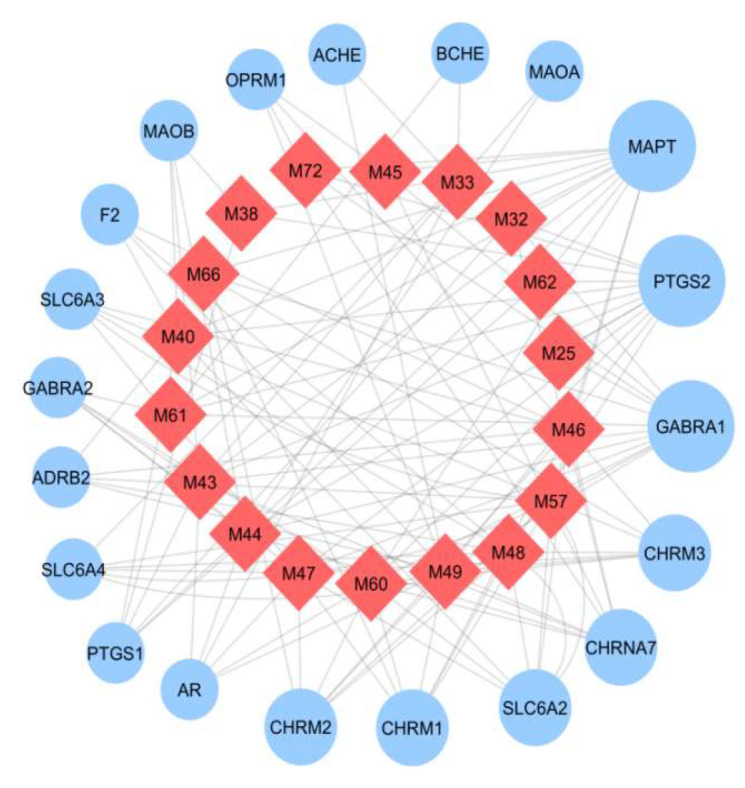
C-T network. The light red diamonds are the compounds and the light blue circles are the targets. The difference in size shows the number of compounds to which the target is connected.

**Figure 7 molecules-27-08294-f007:**
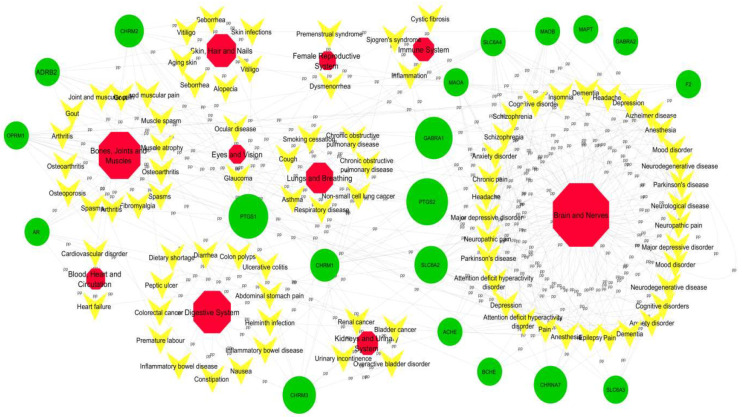
T-D network. Green 20 circles represent the targets of Baizhu. The 10 red octagons represent systems and 78 yellow “V” s represent diseases.

**Figure 8 molecules-27-08294-f008:**
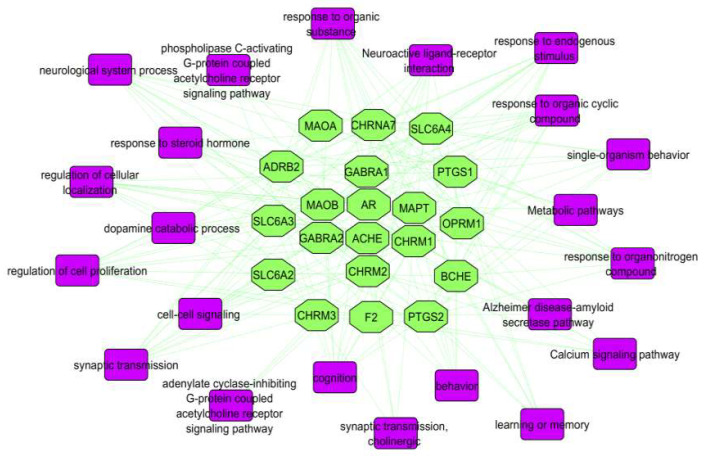
T-P network. Green hexagonals represent targets and purple rectangles represent pathways.

**Figure 9 molecules-27-08294-f009:**
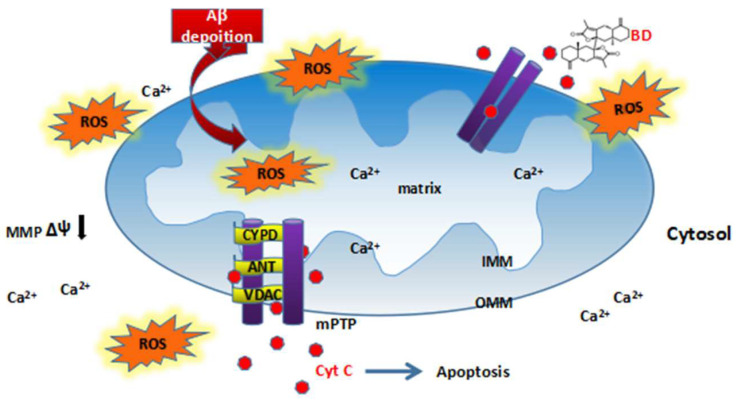
Proposed mechanism of action of biatractylolide (BD). Aβ deposition causes reactive oxygen species (ROS) production, which leads to a change in mitochondrial membrane potential (MMP), opening of mitochondrial permeability transition pore (mPTP), allowing release of CytC (apoptotic factor) into cytosol. BD decreases the production of ROS and inhibits the opening of mPTP. IMM, inner membrane; OMM, outer membrane.

**Figure 10 molecules-27-08294-f010:**
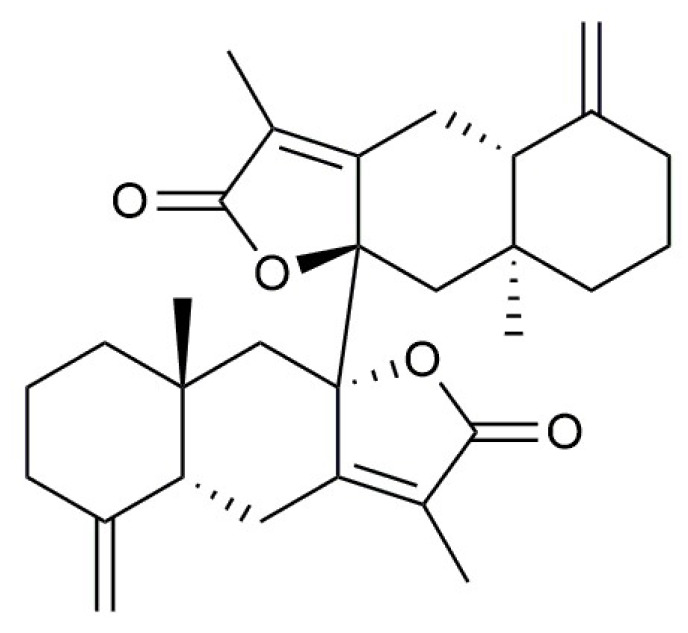
Chemical structure of biatractylolide (BD). Molecular formula: C_30_H_38_O_4_, molecular weight: 462.63 g/mol.

## Data Availability

Not applicable.

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
