# Peer review of "Neuroprotective Effects of the Psychoactive Compound Biatractylolide (BD) in Alzheimer’s Disease"

_molecules, 2022, doi:10.3390/molecules27238294_

Round 1

Reviewer 1 Report

Overall this is an OK manuscript for Molecules. I have some comments for this manuscript. as bellow

1) I am not quite clear the purpose of the compound-target (C-T) and target-disease (T-D) networks in this story? The authors screened 55 compounds (which I could not know what these compounds are currently) but they only test 1 compound.

2) BD is such a important target compound in the story. From the structure the authors posted it seems like it was optically pure, Does the Baizhu only contain this chiral type BD? And, Is the structure of BD the authors posted correct? Please check the chirality of the carbon nearest to the vinyl double bond.

3) I assume that μm is different from μM, please double check the fig 3

4) I also have some concerns in fig 4, (c), I do not think there are too much different in 5, 10 and 20μM . (h), 20μM equal to control?

Author Response

Dear reviewer:
Thank you for your decision and constructive comments on my manuscript. We have carefully considered the suggestion of Reviewer and make some changes. We have tried our best to improve and made some changes in the manuscript.The following is a point-by-point response to all comments.

Question 1: I am not quite clear the purpose of the compound-target (C-T) and target-disease (T-D) networks in this story? The authors screened 55 compounds (which I could not know what

these compounds are currently) but they only test 1 compound

Response: Thank you very much for your constructive questions. Compound-target (C-T) and target-disease (T-D) networks were used to identify potential drugs and their corresponding mechanisms based on the compound-target-disease. The C-T network was generated by linking the screened candidate compounds and their potential targets. The T-D network was constructed by relating relevant targets to their diseases. We have used C-T and T-D network pharmacology studies to detect pharmacologically active compounds in the components of the traditional Chinese herb Atractylodes macrocephala and to predict the potential targets of Atractylodes macrocephala and the targets of related diseases. The compound of traditional Chinese medicine atractylodes is complex. Biatractylolide is a new compound of atractylodes and has research value. And our group has long been devoted to the research of Biatractylolide and has long pursued the mechanism of Biatractylolide against Alzheimer's disease. This time we have focused on mPTP for our research. Other mechanisms can be investigated in other articles. (see references below)

  • Xie YC, Ning N, Zhu L, Li DN, Feng X, Yang XP. Primary Investigation for the Mechanism of Biatractylolide from Atractylodis Macrocephalae Rhizoma as an Acetylcholinesterase Inhibitor. Evid Based Complement Alternat Med. 2016;2016:7481323. doi: 10.1155/2016/7481323. Epub 2016 Aug 24. PMID: 27642355; PMCID: PMC5013199.
  • 冯星,王正濂,林永成,周源,刘英姿,杨华中.双白术内酯对Aβ1-40致痴呆模型大鼠的作用[J].中国药理学通报,2009,25(07):949-951.

Question 2: BD is such a important target compound in the story. From the structure the authors posted it seems like it was optically pure, Does the Baizhu only contain this chiral type BD? And, Is the structure of BD the authors posted correct? Please check the chirality of the carbon nearest to the vinyl double bond.

Response: Biatractylolide is a chiral compound in Atractylodes macrocephala. Due to the diverse and complex chemical composition of Atractylodes, there are other chiral compounds present. Thank you for reviewing the structure of our Atractylenolide compound. We have double-checked that the structural formula of Biatractylolide is correct. (see references below)

(1)Xie YC, Ning N, Zhu L, Li DN, Feng X, Yang XP. Primary Investigation for the Mechanism of Biatractylolide from Atractylodis Macrocephalae Rhizoma as an Acetylcholinesterase Inhibitor. Evid Based Complement Alternat Med. 2016;2016:7481323. doi: 10.1155/2016/7481323. Epub 2016 Aug 24. PMID: 27642355; PMCID: PMC5013199.

Question 3: I assume that μm is different from μM, please double check the fig 3

Response: Thank you for your helpful comments, we have revised the concentration units in Figure 3 to the correct form. We have also checked the full text again to ensure that nothing else has been missed.

Question 4: I also have some concerns in fig 4, (c), I do not think there are too much different in 5, 10 and 20μM. (h), equal to control?

Response: Thank you for your constructive suggestions. Due to an error in collating a large amount of data, we have incorrectly placed fig 4, (a), (c). After careful examination we have placed the correct image in fig 4, (a), (c) and shown that compound BD reduces the increase in CypD induced by Aβ25-35 and that this data is statistically significant. Fig 4, (h) was incorrectly labelled and has been corrected and marked on the figure. The results showed that BD could significantly increase the CytC in mitochondria compared with Aβ group, and the data was statistically significant In the meantime we have checked the whole text again to avoid such errors.

Reviewer 2 Report

Manuscript Title: Neuroprotective effects of the psychoactive compound biatractylolide (BD) in Alzheimer's disease  

Manuscript Number: molecules-2042606

Article Type: Article

Comments:

The manuscript “Neuroprotective effects of the psychoactive compound biatractylolide (BD) in Alzheimer's disease” by Xing Feng and co-workers used the Biatractylolideis a kind of internal symmetry double sesquiterpene novel ester compound isolated from the Chinese medicinal plant Baizhu, has neuroprotective effects in Alzheimer's disease. The methods were well developed a systematic pharmacological model based on chemical pharmacokinetic and pharmacological data to identify potential compounds and targets of Baizhu. The authors assayed the neuroprotective effects of biatractylolide (BD) on PC12 (rat adrenal pheochromocytoma cells) and SH-SY5Y (human bone marrow neuroblastoma cells) by in vitro experiments. The results of in vitro experiments that biatractylolide (BD) could inhibit Aβ by reducing oxidative stress and decreasing CytC release-induced mPTP opening. This study validated biatractylolide (BD) as a promising novel compound for the treatment of Alzheimer's disease.   

I have gone through the whole manuscript and the authors drafted the manuscript with supporting data generation & interpretation, and reference citations. The authors need to be revised more corrections in the attached draft. Please find the yellow color highlights and added comments in the attached pdf file. I have found so many grammatical errors and don’t see the proper spacing between the words and references citation. Please put proper attention while writing peer-reviewed journals. However, the authors provided some valuable work that could be interesting. After careful reading, I am considering this manuscript needs a minor revision. Please re-submit the manuscript with the improved and revised version.

Modest items that need to be addressed include:

1.       The abstract is not convincing to me and please follow the instructions and the guidelines of the ‘Molecules’. Please remove the ‘(1) Background:, (2) Methods:, and (3) Results:’ highlighted part and write in a sentence format and authors must be included some key contributions and output from the work accordingly,. 

2.       Manuscript English grammar needs to be corrected. (Please find some of the grammar corrections in the attached manuscript).

3.       Please merge paragraphs 2 and 3 into a single paragraph.

4.       Can you give me the justification for the values of MTT assay ‘Aβ25-35, PC12 and SH-SY5Y cells proliferated in a dose-dependent 87 manner, with 77.8 ± 1.5%,84.9 ± 1.6 %, 90.5 ± 0.3% and SH-SY5Y 71.1± 0.4 %, 77.1 ± 04 %, 88 82.2 ± 1.4 % cell lines’. The interpretation of values not convincing to me please elaborate on the results and describe them appropriately.

5.  Measurement of intracellular ROS, pre-treatment with BD at 5, 10, 20 uM, the relative fluorescence intensity of ROS to 149.5 ±4.5%, 118.5 ±2.5%, 110 112.5 ± 6.5% in PC12 cells. The values exceed more than 100%. Can you comment on this?

Author Response

Dear reviewer:
Thank you for your decision and constructive comments on my manuscript. We have carefully considered the suggestion of Reviewer and make some changes. We have tried our best to improve and made some changes in the manuscript. The following is a point-by-point response to all comments.

Question 1. The abstract is not convincing to me and please follow the instructions and the guidelines of the ‘Molecules’. Please remove the ‘(1) Background, (2) Methods:, and (3) Results:’ highlighted part and write in a sentence format and authors must be included some key contributions and output from the work accordingly,.

Response: Thank you for your helpful suggestions. We have re-written this part according to the Reviewer's suggestion and the requirements of Molecules. The highlighted part of the file you sent us has been completely revised. And for the author contribution part, we asked all authors to agree with this ‘author contribution’.

Question 2: Manuscript English grammar needs to be corrected. (Please find some of the grammar corrections in the attached manuscript).

Response: Thank you for your valuable comment, we apologize for the poor language of our manuscript. We have carefully checked the grammar in the manuscript and made corrections. Revision marks have been left in the sections of the article that have been revised. We hope that the language of this article has improved substantially after this revision.

Question 3.  Please merge paragraphs 2 and 3 into a single paragraph.

Response: It is really true as Reviewer suggested that we have revised the second and third paragraphs in the manuscript by combining them into one paragraph, as you can see in the highlighted section of the revised version of the manuscript that we have uploaded.

Question 4. Can you give me the justification for the values of MTT assay ‘Aβ25-35, PC12 and SH-SY5Y cells proliferated in a dose-dependent 87 manner, with 77.8 ± 1.5%,84.9 ± 1.6 %, 90.5 ± 0.3% and SH-SY5Y 71.1± 0.4 %, 77.1 ± 04 %, 88 82.2 ± 1.4 % cell lines’. The interpretation of values not convincing to me please elaborate on the results and describe them appropriately.

Response: Thank you very much for your helpful comments. This MTT result indicates that the cell viability of SH-SY5Y cells and PC12 cells was significantly reduced after treatment with Aβ25-35. However, the survival of SH-SY5Y cells was significantly increased after pretreatment with BD (5 μM, 10 μM and 20 μM respectively). The best result was an increase in cell survival to 90.5 ± 0.3% at a BD concentration of 20 μM. The survival rate of PC12 cells was also significantly increased after pretreatment with BD (5 μM, 10 μM and 20 μM, respectively), with the cell survival rate increasing to 82.2 ± 1.4 % at a BD concentration of 20 μM. This result suggests that BD has a significant effect on improving cell viability in both cell.

Question 5. Measurement of intracellular ROS, pre-treatment with BD at 5, 10, 20 uM, the relative fluorescence intensity of ROS to 149.5 ±4.5%, 118.5 ±2.5%, 110 112.5 ± 6.5% in PC12 cells. The values exceed more than 100%. Can you comment on this?

Response: Thank you for your valuable comment. The ROS assay compared the Aβ25-35 group and the drug pretreatment group with the control group, respectively.ROS was significantly higher in the cells after Aβ25-35 treatment. In the cells pretreated with BD, the protective effect of BD reduced the increase in ROS caused by Aβ25-35, and the ROS in the cells was significantly lower but at a slightly higher level than in the control group. Therefore the values exceed more than 100%. This result indicates that BD has the effect of reducing Aβ25-35-induced ROS increased.
